# Biophysical mechanism underlying compensatory preservation of neural synchrony over the adult lifespan

Anagh Pathak [1]✉, Vivek Sharma[1], Dipanjan Roy[2] & Arpan Banerjee [1]✉

We propose that the preservation of functional integration, estimated from measures of neural synchrony, is a key objective of neurocompensatory mechanisms associated with healthy human ageing. To support this proposal, we demonstrate how phase-locking at the peak alpha frequency in Magnetoencephalography recordings remains invariant over the lifespan in a large cohort of human participants, aged 18-88 years. Using empirically derived connection topologies from diffusion tensor imaging data, we create an in-silico model of whole-brain alpha dynamics. We show that enhancing inter-areal coupling can cancel the effect of increased axonal transmission delays associated with age-related degeneration of white matter tracts, albeit at slower network frequencies. By deriving analytical solutions for simplified connection topologies, we further establish the theoretical principles underlying compensatory network re-organization. Our findings suggest that frequency slowing with age- frequently observed in the alpha band in diverse populations- may be viewed as an epiphenomenon of the underlying compensatory mechanism.

[1] National Brain Research Centre, Manesar, Gurgaon, Haryana, India. [2] Center for Brain Research and Applications, School of AIDE, IIT Jodhpur, Karwar, Rajasthan, India. ✉email: anagh.b16@nbrc.ac.in; arpan@nbrc.ac.in

Compensatory mechanisms play a central role in the maintenance of complex biological systems like the brain. The basic principle underlying compensatory processes is that similar activity patterns in biological networks can be achieved by multiple underlying dynamical configurations of the system[1,2]. For example, neural networks can preserve circuit operation around a target set point in the face of continuous molecular turnover by undergoing synaptic modifications[2]. Although compensation has been mostly studied at fast time-scales, compensatory processes are also relevant at ultra-slow timescales such as lifespan ageing[3,4]. A persistent debate in the field of ageing neuroscience centers around the question of whether functional neuromarkers of healthy ageing indicate a gradual degradation of brain structure or, the presence of compensatory reorganization mechanisms that counteract the dele-terious effects of structural loss[4–6]. Compensatory theories posit that certain features of brain ageing, for example, enhanced activation in specific brain regions with age are markers of compensation rather than functional decline[4,7]. Behaviorally, compensatory theories of ageing are further supported by reports of age-related maintenance of cognitive domains such as language comprehension and crystallized intelligence[8,9]. However, the task of classifying various markers of brain ageing as either adverse or compensatory is made difficult by the enormous complexity of brain dynamics[3,10]. Biophysically inspired computational models provide crucial mechanistic insights when purely experimental observations are insufficient in resolving competing hypotheses. Therefore, the goal of this paper is twofold—first, to identify dynamical markers that remain invariant with age and second, to elucidate the operational principles that support such invariant relationships in the face of age-related structural decay.

Electro/magneto-encephalographic (EEG/MEG) signals unfolding at millisecond time scales carry signatures of circuit-level neural information processing and serve as an obvious choice for studying the reorganization of brain dynamics over lifespan[11]. EEG/MEG recordings at rest are marked by prominent oscillatory activity in the alpha frequency range (8–12 Hz)[12]. Numerous EEG/MEG studies have shown that the frequency corresponding to the peak power in the alpha band (Peak Alpha Frequency or PAF), reduces with age[13–18]. Brain-wide alpha activity is coordinated by and propagates along white-matter fibers that connect spatially distant brain regions[19–21]. Accordingly, various studies have identified white-matter as a potential locus for resting-state alpha disruption[19–23]. Two obvious questions emerge—is PAF slowing over lifespan mediated by white-matter structural decay and whether frequency slowing plays a compensatory role in overall circuit maintenance.

White-matter fibers consist of myelinated axons which undergo multiple cycles of repair throughout normal ageing[24–27]. However, axonal conduction speeds are only partially restored by remyelination, as remyelinated axons possess shorter internodes as compared to developmentally myelinated axons[26,28,29]. Reduced conduction speeds along white-matter tracts predict slower network frequencies and impaired synchronization in network models of large-scale brain dynamics[30–32]. Left unchecked, progressive reduction in conduction velocity with age may lead to a complete breakdown of synchrony in crucial brain circuits that subserve normal cognitive processes[21,33]. Hence, from a systems-level view, there must exist compensatory mechanisms that arrest functional degeneration.

Here, we hypothesize that PAF slowing with age is sympto-matic of compensation, rather than just being a passive fallout of age-related structural deterioration. To test our claim, we evaluate the degree of neural coordination using phase synchrony measures[34] at different regimes of alpha frequency band in resting-state MEG recordings collected from participants aged 18–88 at the Cambridge Center for Ageing and Neuroscience (Cam-CAN)[35]. Phase synchrony was measured by estimating Phase Locking Value (PLV)—a measure that quantifies the con-sistency of phase relationships between oscillators over a period of time[36–38]. For each subject in the cohort we calculated PLV at three different frequency bands- Lower Alpha (LA, 6–10 Hz), Upper Alpha (UA, 10–14 Hz) and another subject specific band obtained by considering a 4 Hz band centered at the PAF (Subject Specific Alpha or SSA). Several cross-sectional studies have found that spectral features of on-going alpha oscillations exhibit differ-ential effects with age in the lower and higher alpha sub-bands[39–42]. Moreover, the lower and higher alpha bands have been found to be functionally independent in several cognitive and perceptual paradigms[43–45], further justifying a sub-division of the alpha band. We further reasoned that the polarity of correlation between phase coupling and age (whether increasing or decreasing) in each of the three bands would aid model building by further restricting the search space of candidate mechanisms.

Seeking mechanistic insights into age-related compensation, we investigate the relationship between slowing of PAF and phase-locking by constructing an in silico whole-brain model (WBM) of neural coordination. The WBM consisted of coupled differential equations, modeling the phase of autonomous alpha oscillators (Kuramoto model)[46] at nodes chosen from standardized anato-mical parcellations of the human brain[47]. White-matter proper-ties obtained from diffusion tensor imaging (DTI) data, namely—inter-areal connection strength and transmission delays were varied to study the relationship of steady-state phase-locking and network frequency in the alpha band. The complex network obtained from DTI derived topology and a mathematically tractable all-to-all network captured the emergence of a complex interplay between transmission delays and neural coupling in determining phase locking and dominant network frequency.

## Results

We performed MEG data analysis in conjunction with compu-tational modeling to elucidate the relationship of neural coordi-nation associated with PAF and reorganization of white-matter with age ($N = 627$, 315 Females). First, we estimate phase locking —a widely used measure of neural coordination among simul-taneously recorded brain signals (Fig. 1), from eyes-closed rest-ing-state MEG data (8 min/participant). Second, we theoretically demonstrate compensatory reorganization in a simplified net-work model of interacting alpha oscillators connected via all-to-all connection topology. Finally, we simulate the neural dynamics generated by the cortical network spanning the entire brain whose nodes are connected by DTI based structural connectivity with realistic cortical conduction speed to gain insights into the biological mechanisms that guide the slowing of PAF and maintenance of neural coordination.

**Alpha phase locking is preserved at the PAF across age.** MEG recordings from the Cam-CAN lifespan cohort (age range 18–88 years) were used to characterize the relationship of network fre-quency and phase coupling across age. Resting-state MEG data was first pre-processed and then submitted to a source recon-struction pipeline as implemented in the Brainstorm toolbox[48]. Source time series were obtained for predefined anatomical par-cellations (Region of Interest or ROI) in accordance with the Desikan-Killiany parcellation scheme consisting of 68 brain regions[49]. For each subject, power spectra was computed between 1 and 40 Hz for each ROI using the Welch Method (see Meth-ods). For each spectra, background fluctuations ($1/f$ component) were separately modeled using an automated algorithm[50] for the identification of individual peak alpha frequency (see Methods). This was deemed necessary because $1/f$ features in EEG have been

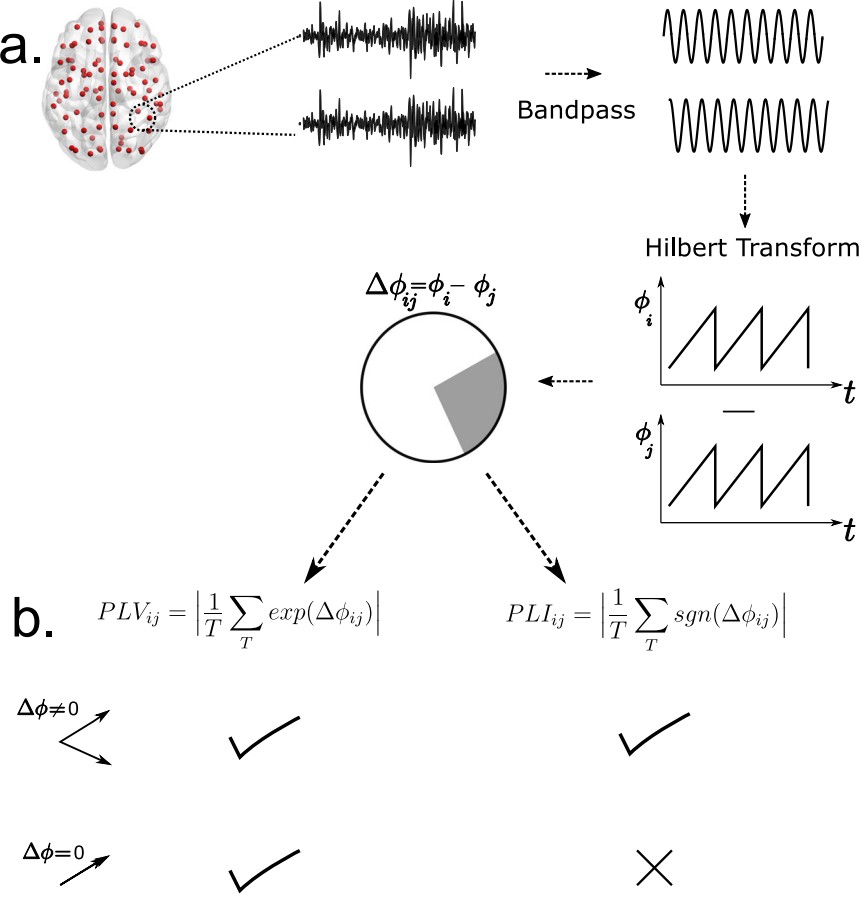

**Fig. 1 Pipeline for estimation of phase locking via PLV and PLI. a** Source-localized MEG signals are bandpass filtered to extract signals in specific frequency bands. Hilbert transform is used to extract instantaneous phase time series. Phase vectors at each timepoint are projected onto a unit circle. **b** Mathematical expressions for the estimation of PLV and PLI. PLV measures zero-phase lags whereas PLI discounts zero-phase lags. This property makes PLI resilient to volume conduction/field spread artifacts that manifest as zero-phase lag correlations.

shown to vary with age[51] and could possibly confound the subsequent analysis. PAF estimated from ROIs were averaged to yield a mean PAF for each subject. Mean PAF was found to significantly reduce with age ($r = -0.38$, $p < 0.0001$) (Fig. 2), similar to patterns reported in earlier studies[14,18].

Next, for each participant we computed the Phase Locking Value (PLV) at the individual PAF between all ROI pairs. Since this band is derived from the PAF estimated separately for each subject, we refer to it as the Subject Specific Alpha (SSA) band. PLV is estimated by bandpass filtering the time series in the frequency band of interest and then using Hilbert transform to obtain the corresponding phase time series. Next, for each ROI pair, phase time series are subtracted to yield phase differences which are then used to estimate consistent phase relations across time (see Methods for mathematical expression). While robust at estimating phase locking, PLV is susceptible to picking up phase correlations that may arise due to the spread of magnetic fields from one brain area to another (known as field spread). Phase correlations due to field spread are most likely to occur at zero-lags. Therefore, in addition to PLV, another measure of phase locking—Phase Lag Index (PLI), which discounts zero-lag correlations, was estimated for each participant and frequency band (see Methods and Fig. 1).

Estimated PLVs/PLIs were averaged across all ROI pairs to obtain one mean PLV/PLI for each participant. For comparison, we also estimated PLV and PLI at two other frequency bands—Lower Alpha (LA, 6–10 Hz) and Upper Alpha (UA, 10–14 Hz). Pearson's linear correlation analysis between age and band-specific phase

locking revealed that both PLV and PLI increased with age in LA band ($r_{PLV} = 0.32$, $p_{PLV} < 0.0001$, $r_{PLI} = 0.21$, $p_{PLI} < 0.0001$); PLV and PLI decreased with age in the UA band ($r_{PLV} = -0.18$, $p_{PLV} < 0.0001$, $r_{PLI} = -0.38$, $p_{PLI} < 0.0001$). In contrast, PLV only marginally increased ($r_{PLV} = 0.1$, $p_{PLV} = 0.03$) while PLI remained invariant with age in the SSA band ($r_{PLI} = 0.01$, $p_{PLI} = 0.5$) (Fig. 2). Surrogate distributions, obtained by randomly shuffling resting-state epochs indicated that PLV and PLI values in the SSA band were significantly higher than what would be expected by chance ($p < 0.01$) (Fig. 2). Taken together, the results indicate that phase locking is preserved at the SSA while the PAF slows with age (Fig. 2). Findings were also replicated at the sensor level ($N = 650$, see Supplementary Fig. 3).

**Conduction delays and coupling modulate network frequency and synchrony in an idealized neural network with all-to-all connections.** We motivate a theoretical understanding of how oscillatory frequency and network synchronization are modulated via connection properties by considering a network of N, Kuramoto phase-oscillators[46]. Oscillators interact with one another according to the following equation-

$$\dot{\theta}_i = \omega_i + \frac{K}{N}\sum_{j=1}^{N}\sin\left(\theta_j(t-\tau) - \theta_i\right) + d\zeta(t) \qquad (1)$$

where, $\theta$ and $\omega$ are the phase and natural frequency of each oscillator. $K$ and $\tau$ specify average coupling strength and transmission delay between any two nodes respectively. Natural

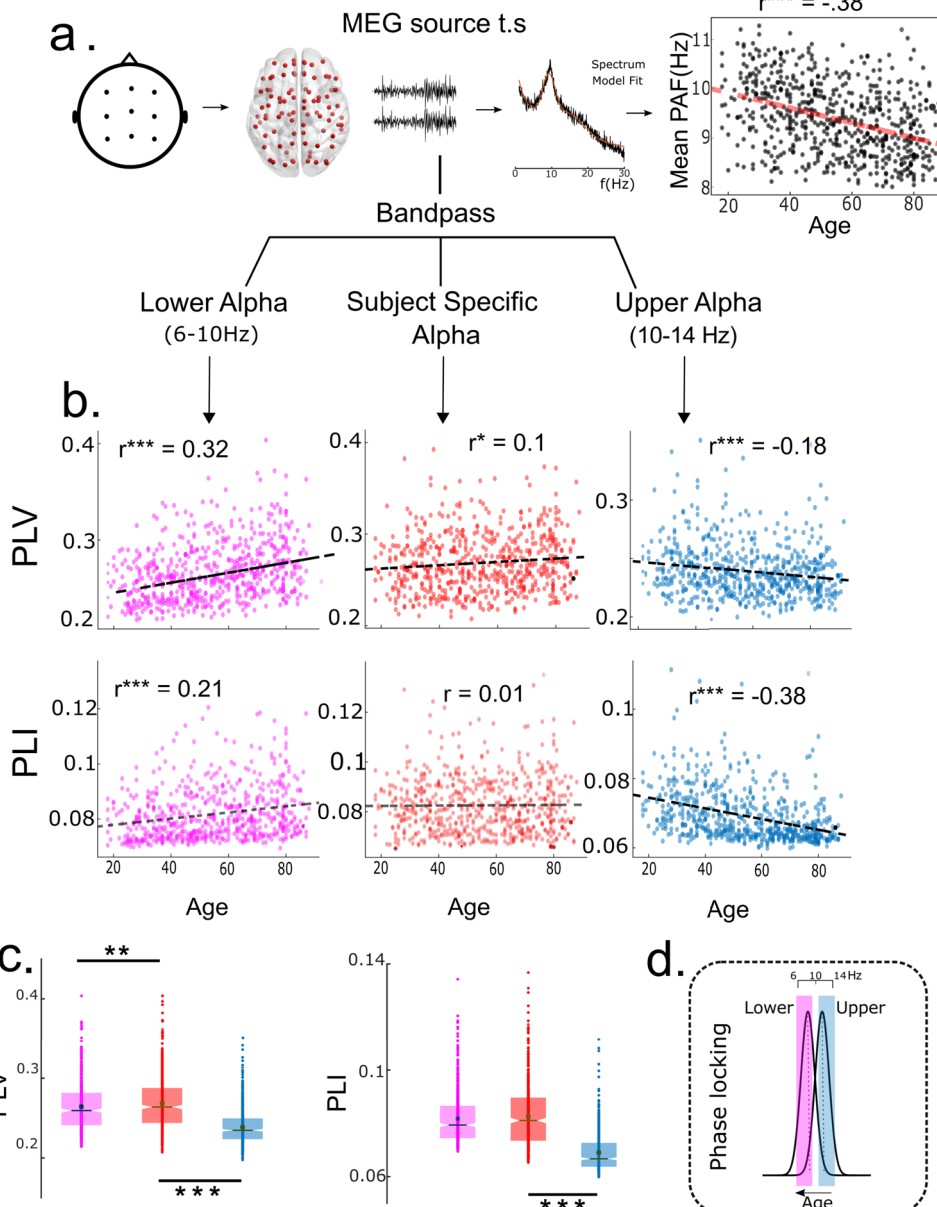

**Fig. 2 Phase locking in the alpha band. a** Overview of the analysis pipeline: rsMEG sensor space data was projected to source space. PSD for each ROI was extracted using Welch method and modeled as a linear superposition of periodic and aperiodic components. Peak frequency was extracted for each brain region and averaged across ROIs to obtain a single mean peak alpha frequency for each subject. Mean peak alpha frequency was found to be negatively correlated with age. Subsequently, phase locking was estimated for each subject using both PLV and PLI. **b** Phase Locking Value (PLV) and Phase Lag Index (PLI) estimated for three frequency bands- LA (6–10 Hz), SSA (PAF − 2 to PAF + 2) and UA (10–14 Hz). **c** PLV and PLI box plot for LA, SSA and UA band, for each box N = 627. Width of the notch is proportional to the interquartile range. Dots represent data points. **d** Schematic: PLV, PLI analysis suggests frequency reorganization that preserves alpha phase locking at reduced peak frequencies.

frequencies are derived from a symmetrical distribution centered at $\mu$. In the most general case, the system is supplied with zero mean Gaussian noise process $\zeta(t)$, with a standard deviation $d$. The network is composed of N oscillators connected according to an all-to-all topology. The order parameter (r) indexes the degree of phase-synchronization in the network

$$z = r(t)e^{i\phi(t)} = \frac{1}{N}\sum_i^N e^{i\theta_i(t)} \qquad (2)$$

such that $r(t) = \left|\frac{1}{N}\sum_i^N e^{i\theta_i(t)}\right|$, with $r = 0$ corresponding to incoherence and $r = 1$ to complete synchronization and $z$ is a complex valued function tracking the global phase synchronization in the network. For smaller coupling values, the incoherent state is stable. The incoherent state loses stability at a critical value of coupling ($K_c$), giving rise to a partially synchronized regime. In the absence of conduction delays ($\tau = 0$), the network of oscillators synchronize at the center frequency ($\mu = 10$ Hz) for $K > K_c$. However, for $\tau \neq 0$, the synchronization frequency ($\Omega$) is different from the center frequency of the distribution of natural frequencies ($\mu$) (Fig. 3)[30]. Specifically, we observe $\Omega < \mu$ for the all-to-all coupled network considered here.

For non-zero delays, the network exhibits multistable states, such that the system can reside in multiple synchronized regimes (see Analytical Solution below), each associated with a different synchronization frequency[30,52]. Heatmap in Fig. 3c, shows the relationship of steady-state collective frequency of the network

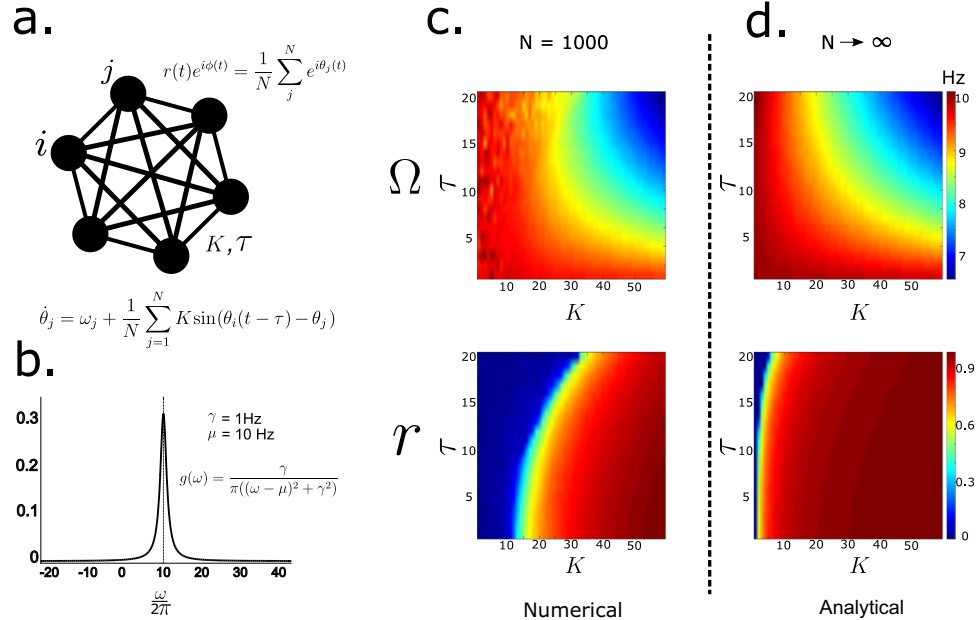

**Fig. 3 Phase dynamics in an idealized network. a** Fully recurrent network of phase oscillators is considered. **b** Natural frequencies of oscillators are drawn from a Lorentzian distribution with, $\gamma = 1\,\text{Hz}$, $\mu = 10\,\text{Hz}$. **c** Steady-state synchronization frequency and order parameter for $N = 1000$ oscillators $d = 0$, obtained by numerical simulations. Delays were varied between 0 and 20 ms. **d** Analytical expressions for synchronization frequency and order parameter derived by reducing the high dimensional system through the Ott-Antonsen method. Network frequency and order parameter are modulated by coupling and delay.

(color) for different $(K, \tau)$. For smaller values of conduction delays, the incoherent network has an average frequency close to the mean of the distribution of natural frequencies ($\mu = 10\,\text{Hz}$). However, for longer delays, the collective synchronization frequency shows considerable suppression.

The order parameter can be shown to evolve via a low-dimensional system of global synchronization manifold under a set of simplifying assumptions[53]. Expressions for steady-state synchronization frequency and order parameter were obtained from the low-dimensional system (see Analytical solution) and compared with parameter space obtained from numerical simulations (1).

**Analytical solution relating synchronization frequency and conduction delays for a reduced system.** We derive analytical expressions for the synchronization frequency ($\Omega$) and steady-state order parameter (r) for the case of a fully recurrent network of phase oscillators ($N \rightarrow \infty$), connected to each other via coupling $K$ subject to delay $\tau$. For simplicity, we consider the noiseless case ($d = 0$). The natural frequencies of oscillators $\omega_j$ are derived from a *Lorentzian* distribution given by

$$g(\omega) = \frac{\gamma}{\pi\left((\omega - \mu)^2 + \gamma^2\right)} \tag{3}$$

Ott and Antonsen[53] showed that the macroscopic dynamics corresponding to Eq. 1 follows a low-dimensional ordinary differential equation (ODE) given by:

$$\dot{z} = (i\mu - \gamma)z - \frac{K}{2}\left(z^2 \overline{z}_{t-\tau} - z_{t-\tau}\right) \tag{4}$$

For details of this step please refer to the Supplementary Material (Supplementary Note 4) or Ott and Antonsen[53]. We require steady-state solutions of the form[53]

$$z = r_0 e^{i\Omega t} \tag{5}$$

where $r_0$ and $\Omega$ are the steady-state order parameter and synchronization frequency respectively. Eqs. 4, 5 lead to

$$
\begin{aligned}
\dot{z} &= r_0 i\Omega e^{i\Omega t} \\
&= (i\mu - \gamma)r_0 e^{i\Omega t} - \frac{K}{2}\left[r_0^2 e^{i2\Omega t} r_0 e^{-i\Omega(t-\tau)} - r_0 e^{i\Omega(t-\tau)}\right]
\end{aligned} \tag{6}
$$

$r_0 = 0$ (incoherent solution) is a trivial solution of Eq. 6 for all $K, \tau, \gamma$. In order to explore coherent solutions we equate the real and imaginary parts on both sides leading to the following transcendental equations

$$\Omega = \mu - K\sin\Omega\tau + \gamma\tan\Omega\tau \tag{7}$$

$$r_0^2 = 1 - \frac{2\gamma}{K\cos\Omega\tau} \tag{8}$$

The requirement $0 \leq r_0^2 \leq 1$ yields the following condition for the existence of coherent solutions

$$\cos\Omega\tau \geq \frac{2\gamma}{K} \tag{9}$$

Equations 7, 8 suggest a mechanism through which frequency and order parameter are modulated as a function of coupling and delay for given $\gamma$ and $\mu$. The transcendental Eq. 7 can be approximately solved by performing Taylor series expansions, $\sin\Omega\tau \rightarrow \Omega\tau - \frac{(\Omega\tau)^3}{3!}$, $\cos\Omega\tau \rightarrow 1 - \frac{(\Omega\tau)^2}{2}$, $\tan\Omega\tau \rightarrow \Omega\tau + \frac{(\Omega\tau)^3}{3}$. Considering the first two terms of the Taylor series the transcendental Eq. 7 can be simplified to

$$(2\gamma + K)\Omega^3\tau^3 + (6\gamma\tau - 6K\tau - 6)\Omega + 6\mu = 0 \tag{10}$$

and the constraint in (9) can be further approximated to

$$\Omega^2\tau^2 \leq 2 - \frac{4\gamma}{K} \tag{11}$$

Subsequently, Eq. 10 is solved numerically by using a MATLAB routine (fsolve) for the parameter space constrained by the

condition (11). The solution shows excellent agreement with numerical solutions with $N = 1000$, as shown in Fig. 3. Most importantly for our study, we find that reductions in network synchrony due to increased conduction delays can be offset by concomitant changes in coupling.

**Reduced network synchrony due to lowered conduction speeds can be rescued by global scaling of connection strength.** In silico modeling of brain dynamics was used to study whether alpha dynamics on white-matter network topology retains the key features exhibited by idealized networks for which analytical relationships between network frequency and neuronal coupling are derived in the previous section. Kuramoto phase-oscillators[46] were placed at anatomical landmarks using the Desikan-Killiany atlas[49]. For specifying white-matter connectivity, we used DTI adjacency matrices from a separate dataset of healthy subjects, as described in Abeysuriya et al.[54]. Conduction delays between all node pairs were estimated by scaling inter-node Euclidean distances by a fixed cortical conduction velocity $v$, $\tau_{ij} = \frac{D_{ij}}{v}$. The biological scenario differs from the idealized network in three key aspects: 1. topology, 2. distribution of natural frequencies and 3. existence of distance dependent delays. Oscillators are set up to interact with one another according to the following equation[46]

$$\dot{\theta}_i = \omega_i + \frac{K}{N} \sum_{j=1}^{N} c_{ij} \sin(\theta_j(t - \tau_{ij}) - \theta_i) + d\zeta(t) \quad (12)$$

where, the term $c_{ij}$ introduces heterogeneous connection weights derived from normalized measures of fiber densities. Similar to the idealized case, network frequency and phase locking values were obtained by varying cortical conduction velocity (v) and global scaling parameter (K). Cortical conduction velocity was varied in the range of 1–30 m/s, in line with previous experimental reports[55]. Altering the conduction velocity changes the distribution of distance dependent transmission delays, whereas changing K is analogous to synaptic scaling. Following[56,57], natural frequencies ($\omega_i$'s) are distributed across ROIs based on node strength ($\omega_{max} = 12$ Hz, $\omega_{min} = 8$ Hz, $\mu = 11.06$ Hz) (Fig. 4b, see Methods for selection of natural frequencies).

Similar to observations in idealized network topology with all-to-all connections, network synchrony is modulated by the combined influence of conduction velocity and global gain parameter (K). Broadly, the system exhibits lower levels of synchrony for very weak coupling and low conduction velocity (Fig. 4). On the other hand, larger coupling and conduction velocity lead to hyper-synchronous states. Somewhere in between lies the partially synchronized state, characterized by high temporal variability in the Kuramoto order parameter. Such maximally metastable regimes are thought to underlie resting-state dynamics[58,59]. Accordingly, we restrict our attention to the metastable regime while considering age-related reduction in conduction velocity. Distribution of metastability values clearly delineates the metastable regime as a distinct mode (Fig. 4e).

To better visualize the relationship of network frequency and phase locking, we plot contour lines. Compensatory balancing of phase locking corresponds to a traversal along the PLV contour lines (Fig. 4c). We observe a robust reduction in mean network frequency along PLV contour lines in the metastable regime (Fig. 4c, d). In contrast, a vertical descent along the y-axis in Fig. 4a, that can be interpreted as a passive decline in conduction velocity without compensation, is not accompanied by any notable reduction in network frequency. This can be gauged by the vertical orientation of frequency contours in the metastable regime.

Interestingly, moving along the PLV contour lines not only preserves network synchrony, but also metastability (red region,

Fig. 4d). Therefore, in addition to preserving mean synchrony levels, synaptic scaling also maintains the temporal richness of alpha dynamics that subserves the dynamic repetoire of core brain areas[60]. A non-compensatory decline in network conduction velocity also raises the possibility of a sudden increase in network frequency owing to a complete breakdown of network synchrony due to high network delays. However, traversing along PLV contour lines assures a monotonic reduction of network frequency, while maintaining synchronicity among brain areas. Model simulations with different noise amplitudes and distributions of natural frequency led to qualitatively similar results (Supplementary Note 1, Supplementary Fig. 1). To demonstrate the generality of our results, we replicated the analysis using a different connectomic dataset and parcellation scheme (automated anatomical labeling, AAL) as described in[61] (Supplementary Note 2, Supplementary Fig. 2).

**Discussion**
In this study we propose that functional integration achieved via neural synchrony is an important neurocompensatory mechanism that is underway during healthy physiological ageing. The key entry point that led us towards this understanding is a widely reported phenomenon—age-related decrease in individual peak alpha frequency[13,14,41,51] which we demonstrate to be the outcome of a dynamic functional compensation process. Compensation preserves network synchrony in response to inclement enhancement in transmission delays stemming from the deterioration of axonal tracts as a function of age. We test our hypothesis on empirical MEG recordings by estimating measures of network synchrony at the PAF in both source (Fig. 2) and sensor levels (Supplementary Note 3, Supplementary Fig. 3). We validate this hypothesis by employing two complementary measures of phase synchrony—Phase Locking Value (PLV) and Phase Lag Index (PLI) on source-reconstructed MEG data made publicly available by Cam-CAN. While PLV estimates consistency of phase differences, PLI additionally controls for volume conduction by discounting zero-lags[62]. Through surrogate testing we confirm that PLV and PLI indicate significant phase relationships and are not artifacts of sample size bias[62]. Our experimental results corroborate the findings of preserved PAF connectivity by Scally et al. which were published for a smaller sample size on EEG sensor level data[41] (Supplementary Note 5). Here we also demonstrate the preservation of network synchrony at PAF in both source and sensor level data, thus confirming and expanding the scope of previous findings[41] (Supplementary Fig. 5). Furthermore, partitioning the alpha band into lower and higher sub-bands offers both practical and mechanistic insights into the nature of dynamical reorganization over the course of physiological ageing. By studying phase coupling separately for LA, UA, and SSA bands we are able to confirm that in the presence of group variability in peak frequencies, the use of predefined frequency bands to characterize group differences in functional connectivity leads to the detection of spurious relationships[41]. Therefore, our findings caution against the use of pre-selected frequency bands for studying cognitive phenomenon[41] while also highlighting the importance of considering inter-subject/condition variability in the distribution of neural oscillations[63,64]. A similar approach of dividing alpha into a lower and higher sub-band is followed by Vecchio et al. who showed that graph-theoretic connectivity in the alpha band decreases with age in the higher alpha sub-band[40]. Gaal et al. also study physiological ageing by performing EEG reactivity analysis by considering two sub-bands within alpha[42]. Similarly Babiloni et al. characterize age trajectories separately for alpha sub-bands and report differential effects in the lower and higher alpha frequency ranges[13].

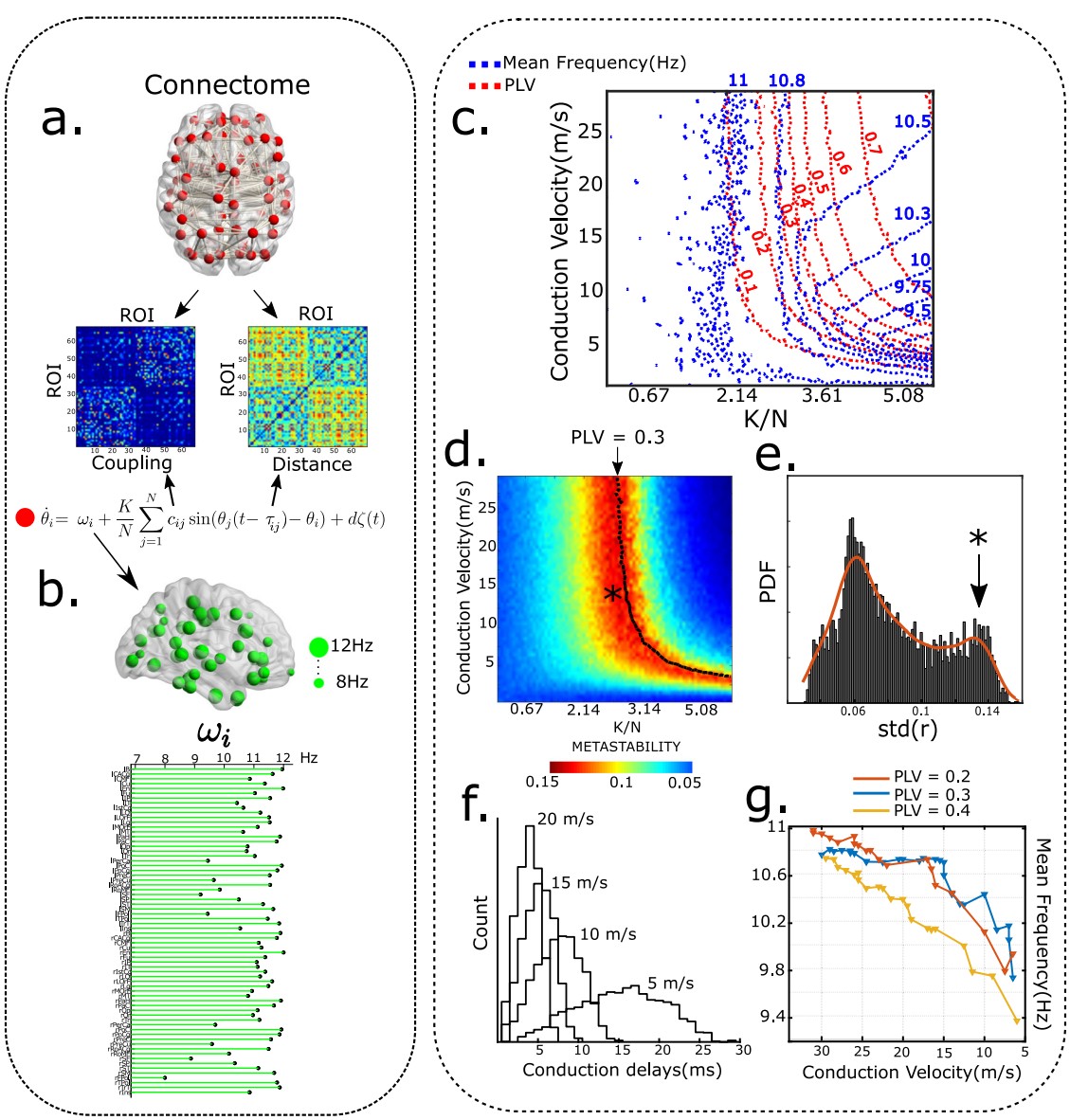

**Fig. 4 Large-scale alpha phase locking. a** Model overview-DTI connectivity and distribution of inter-node distances. Equations governing node dynamics. **b** Distribution of natural frequencies. Green spheres represent magnitude of natural frequency. ROI-wise distribution of natural frequencies. **c** Contour plot indicating isolines for mean frequency (blue) and PLV (red) as a function of global coupling and conduction velocity, Noise amplitude (d) = 3, $\omega_{max} = 12$ Hz, $\omega_{min} = 8$ Hz. PLV and PAF remain constant along isolines. **d** Metastability measured as the standard deviation of the order parameter plotted as a function of conduction velocity and global coupling. Dotted line indicates PLV isoline. Asterisk corresponds to region with maximum metastability. **e** Distribution of metastability in the parameter space. Asterisk in heatmap corresponds to second mode of the gaussian. **f** Distribution of conduction delays (in ms), for conduction velocity = 5, 10, 15.20 m/s. **g** Frequency depression along isolines corresponding to PLV = 0.2, 0.3, and 0.4.

Besides ageing, another group of studies have divided the alpha band to shed light on neurological disorders such as mild cognitive impairment (MCI)[65,66], persistent vegetative state[67] and Autism spectrum disorder (ASD)[68]. Most importantly, age-related trajectories in the LA, UA, and SSA band serve as the explanandum for our subsequent model. While preserved age trajectories in the SSA band may come about due to multiple underlying phenomena, the added constraints imposed by LA and UA trajectories allow us to restrict the computational model to a smaller set of candidate mechanisms (Fig. 2d). In short, our computational model aimed to explain three crucial empirical observations—(1) Preserved phase locking in the SSA band alongside a reduction in PAF, (2) Enhanced phase locking with age in the LA band, and (3) Reduced phase-locking with age in the UA band. We argue that age-associated enhancement in

conduction delays offers a parsimonious explanation for all three empirical findings.

Numerous studies have speculated a prominent role for white-matter fibers in modulating alpha synchrony[20,22,23]. Therefore, in order to study the relationship of network frequency and synchrony, we reduce large-scale white-matter network to its basic dynamical elements- conduction delays and inter-areal coupling that forms the backbone of a whole-brain connectome. Each node in the connectome is considered to be a unit amplitude limit-cycle oscillator (an idealized autonomous oscillator), described by its phase (Fig. 3). Anatomically, each autonomous alpha oscillator can be identified with a self-sustained thalamo-cortical unit, or alternatively, pacemaker populations such as the infragranular and supragranular layer in V2 and V4[69,70]. Both numerical and analytical approaches on idealized network with all-to-all

connections confirm how changes in average conduction delays may be offset by modifications in coupling, and that frequency slowing is a collateral to this compensation. Next, we extend our model to include network topology estimated from empirical human white-matter connectivity and find robust frequency modulation with average conduction speed change and global coupling. We track trajectories in the global coupling-conduction velocity space that preserve phase locking, restricting our attention to the partially synchronized regime. Our results unequivocally demonstrate that frequency slowing emerges as the system attempts to maintain phase locking in response to a reduction in conduction velocity by modulation of inter-areal connectivity (Fig. 4).

As early as the 1950s, Norbert Wiener had hypothesized that independent oscillators with natural frequencies close to 10 Hz interact with one another to shape alpha rhythmicity[71,72]. According to Wiener, alpha activity emerges from frequency pulling between individual alpha oscillators which possess slightly different natural frequencies[71]. This idea is regarded as one of the earliest models of collective dynamics of biological oscillators[72]. In the intervening years, models of collective synchronization have become a mainstay of neuroscience, having been employed to explain diverse phenomenon such as traveling brain waves, fluctuating beta oscillations, fMRI functional connectivity, large-scale brain synchronization, myelin plasticity etc.[32,47,73–75]. In the present article, we adapt Wiener's idea of frequency pulling to explain the gradual slowing of alpha frequency with age. Specifically, we show that frequency pulling in the presence of conduction delays, biases the system to synchronize at lower frequencies. Intuitively, the mechanism proposed here is analogous to a group dance, where complicated dance moves are initially practiced slowly, since it is easier to maintain lockstep at lower speeds. Similarly, upscaling inter-areal coupling (global synaptic scaling) allows for the maintenance of network *lockstep* at slower coordination frequencies.

Similar homeostatic mechanisms that regulate circuit output have been identified elsewhere. For example, neurons in the visual cortex of developing rodents undergo synaptic scaling in response to visual inputs[76]. Synaptic scaling has also been shown to compensate for neuron number variability in the crustacean stomatogastric ganglion[77]. Recently, Santin et al. elegantly demonstrate how respiratory motor neurons in the bullfrog can dynamically regulate breathing by modifying synaptic strengths after long periods of inactivity[78]. From the perspective of communication through coherence (CTC) hypothesis[79], alpha phase-locking constitutes an *information channel*, whereby distant oscillators with slightly different peak frequencies communicate with one another through leader-laggard phase relations. Consistent phase locking, a prerequisite for effective communication across brain regions, entails that oscillators adjust their individual frequencies under the combined influence of coupling and transmission delays. Thus, there emerges a clear relationship between oscillation frequency and phase connectivity. Therefore, our central hypothesis is that frequency shifts with ageing need to be understood in the context of homeostatic maintenance of large-scale phase locking. Understanding the precise mechanism of frequency slowing—whether adverse or compensatory—has far-reaching consequences for characterizing age-associated neuropathologies like Dementia and Alzheimer's disease, which share PAF slowing as a prominent feature[39,80,81]. In the framework proposed here, greater frequency slowing in AD may result from the higher demands placed on compensatory processes by accelerated demyelination. Therefore, our model supports a growing view that suggests a greater role for white-matter abnormality in explaining AD progression[82].

Compensatory models of ageing have been proposed to account for the finding that many individuals continue to function remarkably well with age despite substantial structural loss. For example, according to the scaffolding theory of ageing and cognition (STAC)[4], the ageing brain can preserve cognitive function in the face of age-associated neural changes like volume shrinkage, white-matter degeneration, cortical thinning, and dopamine depletion by recruiting alternate neural pathways, referred as scaffolds. While the STAC model has succeeded in explaining a number of observations in ageing neuroscience at the cognitive level, we still lack a clear understanding of the dynamical principles that facilitate compensation. Thus, our model departs from the standard conceptualization of alpha slowing as an adverse outcome of ageing. Rather, we recast frequency slowing as a tell-tale signature of neural compensation. Earlier models have conceptualized alpha slowing as a passive process, resulting from the gradual decline of system parameters. For example, Da Silva et al.[83] model EEG maturation by using a neural mean field model. Their model consists of two populations of neurons: thalamic and cortical, driven by multiple uncorrelated noise sources. By changing the feedback coupling parameters the authors obtained a family of spectral curves that closely resemble developmental trajectories. However, the model lacks axonal delays, which are known to undergo age-related changes and are held to be major drivers for the evolution of cortical networks. Similarly, Van Albada et al.[84] employ a more detailed neural mean field model of the thalamo-cortical system[85] to investigate age-associated changes in EEG spectral parameters and found white-matter stabilization and regression to be a major determinant of EEG characteristics across lifespan. However, the approach, models the gross EEG spectrum for estimating model parameters, making it hard to dissociate specific mechanisms responsible for age-related changes in narrow-band frequencies. More recently, Bhattacharya et al.[86] used a variant of the Lopes Da Silva model to study slowing of peak alpha oscillations in the context of Alzheimer's disease, implicating thalamic inhibition as the principal driver of alpha slowing, however, as with the original Lopes Da Silva model, this model does not consider axonal conduction delays. Future efforts can build on the model proposed here to tease out how compensatory processes operate in various other contexts, such as rehabilitation from stroke, recovery from traumatic brain injuries to predict recovery timelines and to detect critical times for intervention. Going forward, our model may be augmented by new imaging paradigms (e.g. g-ratio mapping to introduce heterogeneity in conduction speeds[87]).

## Glossary of terms

**Phase locking**. Refers to the tendency of cyclic time series to synchronize due to mutual interactions. For example, two pendulum clocks connected weakly get aligned over time. In neuroscience the *binding by synchrony* hypothesis holds that object attributes are represented in the brain by synchronous activity of neural ensembles. Synchrony may be achieved at zero-lag (no phase difference) or at non-zero-phase lags (constant phase difference).

**Compensation**. The ability of biological systems to adaptively change network parameters in order to maintain circuit/behavioral output. Consequently, identical activity patterns can be seen for networks having widely diverging system parameters. For example, cerebellar Purkinje neurons may display similar electrical activity inspite of having different ratios of sodium and calcium currents.

**Alpha band**. Prominent oscillatory feature of human electro-physiological recordings, characterized by 8–12 oscillatory cycles per second. Enhanced in the eyes-closed condition, alpha oscillations are implicated in diverse cognitive phenomenon like attention and working memory. Alpha oscillations undergo evolution over the course of development and senescence.

**Source reconstruction**. A set of mathematical algorithms that employ electrode/sensor level time series to estimate neural time courses. Sensor level recordings are assumed to be superpositions of multiple neural generators each casting their influence subject to distance from sensors, properties of intervening tissue etc. Contribution of each neural generator is projected onto scalp surface by employing Maxwell's equations (Forward model). Model inversion (inverse problem) yields reconstructed neural time courses.

**Delay differential equation**. A class of differential equation in which the derivative of the state variable depends upon the value of the variable at an earlier time. For example, in aircraft dynamics there exist finite temporal lags between pilot input and movement of control surfaces like rudder and elevators. Biological systems are also replete with temporal lags (e.g., nerve conduction, auditory delay lines etc). In contrast to ordinary differential equations which require a finite set of initial conditions, delay differential equations require the variable to be defined at all points prior to the initial timepoint and the maximum delay, making the system infinite dimensional.

## Methods

**MEG data description**. Data used in the preparation of this work was obtained from the CamCAN repository (available at http://www.mrc-cbu.cam.ac.uk/datasets/camcan/)[35]. Cam-CAN dataset was collected in compliance with the Helsinki Declaration, and has been approved by the local ethics committee, Cambridgeshire 2 Research Ethics Committee (reference:10/H0308/50). Following written informed consent, MEG data was collected using a 306-sensor (102 magnetometers and 204 orthogonal planar magnetometers) VectorView MEG System by Elekta Neuromag, Helsinki, located at the MRC Cognition and Brain Sciences Unit. Data were digitized at 1 kHz with a high pass filter of cutoff 0.03 Hz. Head position was monitored continuously using four Head Position Indicator coils. Horizontal and vertical electrooculogram were recorded using two pairs of bipolar electrodes. One pair of bipolar electrodes were used to record electrocardiogram for pulse-related artifact removal during offline analysis. The data presented here consisted only of resting state, where the subjects sat still with their eyes closed for a minimum duration of 8 min and 40 s.

**MEG preprocessing and source reconstruction**. Brainstorm package was used for processing MEG data[48]. For each participant, MEG recordings (.fif files) were first loaded. Heartbeat and eyeblink events were detected using EOG and ECG recordings through an automated procedure. SSP projectors were applied to remove heartbeat and eyeblink artifacts. Eyeblink projectors could be successfully applied to 627 out of 650 participants; the remaining 23 participants had to be excluded. Time series were then resampled to 100 Hz in order to reduce computational loads for the subsequent analysis. DC offset and linear trends were removed and the signal was bandpass filtered between 1–40 Hz. Noise covariance matrix for each participant was estimated from empty room recordings provided with the Cam-CAN distribution[35]. Head model was computed from a standard anatomical template provided in Brainstorm (ICBM152 MNI). Current density map was estimated in accordance with the Minimum Norm Imaging[88] method for dipoles oriented normally to the cortex using a depth weighting parameter (order = 0.5, MaximalAmount = 10). Noise covariance was regularized with the median eigenvalues. Regularization was implemented with an SNR set to 3. Source time series were projected to 68 brain parcellations according to the Desikan-Killiany atlas[49].

**Peak alpha frequency (PAF) estimation**. All subsequent analysis was performed on the resulting data. ROI-wise power spectral density was estimated using Welch periodogram method[89]. In accordance with the Welch method, the time series was split into 5s windows with 50% overlap. Each window was multiplied with a tapering window (Hanning window) to suppress the contribution of spectral leakage. Discrete fourier transform was then performed to yield fourier coefficients, which were then squared to yield spectral power. Finally, subject spectrum was obtained by averaging across windows. Power spectrum of electrophysiological recordings consist of both periodic and aperiodic components[51,90]. In order to remove the influence of aperiodic $\frac{1}{f}$ component, the spectrum (P) is modeled as:

$$P = L + \sum_{m=0}^{M} G_m \tag{13}$$

$L$, $G_m$ model the aperiodic and periodic components respectively. $G_m$ is approximated as a Gaussian function:

$$G_m = ae^{\frac{-(F-c)^2}{2w^2}} \tag{14}$$

while the aperiodic component ($L$) is modeled in the semi-log power space as:

$$L = b - \log(F^\chi) \tag{15}$$

where, $b$ is an offset and $\chi$ is the slope. $a$, $c$, $w$, $b$, and $\chi$ were estimated through an automated model fitting procedure[50]. Model fitting was performed in the 2–20 Hz range, following Tran et al.[91].

**Phase locked value (PLV) and Phase lag index (PLI)**. Phase locking was estimated by using PLV and PLI measures. Firstly, each 5s epoch for each participant was bandpass filtered in LA, UA, and SSA band. Next, Hilbert transform was performed for each filtered epoch to extract phase time series, $\phi_a(t)$. Phase difference ($\phi_{ab}(t) = \phi_a(t) - \phi_b(t)$) was calculated for each ROI pair. PLV and PLI were estimated as[62]

$$PLV_{ab} = \frac{1}{T}\left|\sum_{t=1}^{T} e^{(i\Delta\phi_{ab}(t))}\right| \tag{16}$$

$$PLI_{ab} = \frac{1}{T}\left|\sum_{t=1}^{T} sign(\Delta\phi_{ab}(t))\right| \tag{17}$$

$$sign = \begin{cases} +1 & \Delta\phi_{ab} > 0 \\ 0 & \Delta\phi_{ab} = 0 \\ -1 & \Delta\phi_{ab} < 0 \end{cases} \tag{18}$$

PLV and PLI were averaged across epochs and ROI pairs.

**Statistics and reproducibility**. Pearson's linear correlation was computed to calculate age-trends for PAF, PLV and PLI ($N = 627$). Surrogate distributions were generated by randomly shuffling variables across 10,000 iterations (Fig. 2). Additionally, bootstrapping was performed to ascertain significant PLV and PLI in the SSA band (threshold = 0.05) for 100 randomly chosen participants. Surrogate distribution for PLV/PLI were obtained by randomly shuffling epochs to produce 100 PLV/PLI values; p-values were estimated from the resulting distribution (see Supplementary Fig. 4). Additional replication of the main results was done using an openly available rsEEG dataset ($N = 111$, see Supplementary Fig. 5).

**Whole-brain model of alpha slowing**. In this study we use the Kuramoto model (1) with conduction delays in order to explain brain-wide slowing of alpha oscillations with age. We demonstrate frequency slowing on two types of topology—(1) fully recurrent, (2) DTI based structural connectivity.

**Structural connectivity (DTI)**. For our study we use pre-processed structural connectivity (SC) matrix derived from Human Connectome Project as provided in Abeysuriya et al.[54]. SC matrices were obtained by performing probabilistic tractography on diffusion MRI data. In short, fiber orientations were calculated from distortion-corrected data, as implemented in FSL. Probtrackx2 was used to detect upto 3 fiber orientations per white-matter voxel. Matrices were reduced to a 68 * 68 scheme, according to the Desikan-Killainy atlas[49]. Adjacency matrices of 40 participants were averaged. Log-transformation was performed to account for algorithmic biases. Conduction delays were obtained by scaling barycentric distances between ROIs by conduction velocity.

**Natural frequency assignment**. Natural frequencies for 68 ROIs were assigned based on anatomical node strengths according to the equation[56,57]

$$\omega_i = \omega_{max} - (\omega_{max} - \omega_{min})\frac{(s_j - s_{min})^2}{(s_{max} - s_{min})^2} \tag{19}$$

Where, $\omega_{max} = 12$ Hz and $\omega_{min} = 8$ Hz are the maximum and minimum oscillatory frequencies, specifying the distribution of alpha frequency across ROIs. $s_{max}$ and $s_{min}$ are the maximum and minimum strengths respectively.

**Metastability**. Metastability refers to the ability of dynamical systems to flexibly engage and disengage without remaining confined in trivial dynamical configurations such as hyper synchrony or incoherence. According to the communication through coherence (CTC) view, brain areas communicate through state dependent phase coupling[58]. In this scheme, resting-state dynamics must display maximal variability in phase configurations (i.e., maximal metastability). The principle of maximal metastability, combined with realistic values of cortical conduction speeds allow us to demarcate relevant regions for exploration in the conduction speed-coupling space. The standard deviation of the order parameter (2) is regarded as a proxy for metastability[58].

**Numerical integration**. For the recurrent network, system of equations represented by (1) was numerically solved for $N = 1000$ oscillators using the Euler method. Integration time step was kept at $dt = 0.001$ s. For DTI connectivity conduction delays were assumed to be integer multiples of $dt$ to avoid use of computationally intensive interpolation schemes. Noise was supplied to each node by multiplying random normal numbers by the noise amplitude, scaled by $\sqrt{dt}$. Each simulation was run for 30 s and first 10 s were discarded and all subsequent analysis was performed with the resulting signal. Neuroscience gateway platform was used to simulate computationally intensive parameter sweeps[92].

**Reporting summary**. Further information on research design is available in the Nature Research Reporting Summary linked to this article.

## Data availability

Data used in the preparation of this work were obtained from the CamCAN repository (available at http://www.mrc-cbu.cam.ac.uk/datasets/camcan/). DTI structural connectivity matrices were downloaded from https://github.com/OHBA-analysis/abeysuriya_wc_isp/tree/master/data_files and https://github.com/juanitacabral/NetworkModel_Toolbox respectively. Data used for plotting Figs. 2 and 4 can be found in Supplementary Data 1.

## Code availability

The codes used for data analyses and modeling along with corresponding documentation are available at https://bitbucket.org/cbdl/wbm_kuramotodelay/src/master/. Codes may also be accessed from Zenodo (DOI: 10.5281/zenodo.6542486).

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

## Acknowledgements

We acknowledge the generous support of NBRC Core funds and the Computing facility. For simulations, resources from Neuroscience Gateway[92] were used. Data collection and sharing for this project was provided by the Cambridge Center for Ageing and Neuroscience (Cam-CAN). Cam-CAN was supported by the UK Biotechnology and Biological Sciences Research Council (Grant BB/H008217/1), together with support from the UK Medical Research Council and University of Cambridge, UK. Dipanjan Roy was supported by Ramalingaswami Fellowship, Department of Biotechnology, Government of India, Award ID: BT/RLF/Re-entry/07/2014 and Department of Science and Technology (DST), Ministry of Science and Technology, Government of India, Award ID: SR/CSRI/21/2016. Arpan Banerjee was supported by Ministry of Youth Affairs and Sports, Government of India, Award ID: F.NO.K-15015/42/2018/SP-V and NBRC Flagship program, Department of Biotechnology, Government of India, Award ID: BT/MED-III/NBRC/Flagship/Flagship2019.

## Author contributions

The study was conceptualized by A.P. and A.B. Methodology was implemented by A.P., V.S., D.R., and A.B. Investigation was carried out by A.P. and A.B. Supervision for the project was provided by D.R. and A.B. Original draft was written by A.P. and A.B. Subsequent drafts were reviewed by D.R. and A.B.

## Competing interests

The authors declare no competing interests.
