## [Peer Review File · Communications Biology]

Reviewers' comments:

Reviewer #1 (Remarks to the Author):

This manuscript by Pathak and others reported an analysis of diffusion MRI and MEG data over the lifespan, aiming to establish a theory explaining slowing down of peak alpha frequency with age. This is an interesting goal and authors are aiming to perform ambitious study. However, I think that this manuscript does not fit well with this journal. I will explain my concerns below.

First, the manuscript is written solely from an engineering standpoint, the methods are complicated, and there is a lot of assumed knowledge that I suspect readers, and those who are interested in neural oscillation, white matter and aging, will not have. If authors aim to publish this work in general biology/neuroscience journals, an ample revision is necessary to contextualize and relate the description with existing neuroscience knowledge. Important concepts, which are necessary to understand a scope of this work, must be explained in more detail.

For example, many acronyms (PAF, PLI, PLV, or IPAF) were used in this manuscript but they are not carefully described to readers who are not very familiar with EEG/MEG. It seems that authors assume that readers have prior knowledge on these terms, but I don't think that all readers have this knowledge considering the scope of this journal. Importantly, some of acronyms, such as IPAF, were used without even providing the full terms when they first appear in text. For this reason, as a reader I have not chance to fully understand author's method. In addition, Level of details provided regarding methods in Results section should be revised so that it is appropriate for the format with Results placed before Methods. In a current form, reader must go back and forth between Results and Methods to understand what was done.

Authors separately analyzed lower alpha (LA) and upper alpha (UA). However, it is fully unclear to me why authors divided frequency band in this way. Are there any theoretical grounds that these frequency bands have distinct physiological origin? Without providing sufficient context, analysis dividing LA and UA is fully incomprehensible to readers.

Authors averaged peak alpha data across all ROIs. What are the theoretical grounds for this step? Alpha oscillation in occipital cortex and frontal cortex may have very different nature since they are distinct signals and have different functional properties. Averaging peak alpha from all ROIs makes it very complicated to understand the relationship with white matter tracts, which connect specific brain regions.

Information regarding human subjects is not sufficient for readers to understand this work. Even though the analysis was performed by using public datasets, information of human subjects (age, sex, and others) should be described in more detail. Importantly, it is not clear what is the rationale for randomly picking 200 subjects. Why 200 subjects were chosen for this analysis. If authors pick different 200 subjects, will they obtain consistent results?

Since not all readers are fully familiar with MEG analysis pipeline, terms like "boundary element method", "Welch periodogram method" and "Hanning window" should be explained in more transparent way.

It would be useful to have the values of the IPAF centred band as for LA and UA. By looking at Fig 2. a, it seems that they might coincide with LA for some of the age groups?

There are many errors in the manuscript which makes it difficult for me to evaluate a quality of this work. For example:

- Insert a space between a word and the first of the following pair of brackets
- Line 77: typo "deyterminants"  "determinants"
- Line 78: typo "metstability"  "metastability"
- Line 82: typo "was"  "were"
- Line 97: inconsistent capitalisation "Phase locking value"  "Phase Locking Value"
- Line 101: PLI was introduced as phase lagged index in Introduction, not as "Phase Locking

Index”?

- Line 136: inconsistent capitalisation “analytical solution”  “Analytical solution”
- Line 156: ODE not previously used acronym
- Line 381: What is SSA band?
- Line 356: MRC-CBSU not previously used acronym
- Line 362: inconsistent capitalisation “Data selection and Source Reconstruction”  “Data selection and source reconstruction”
- Line 365: inconsistent capitalisation “Middle late”  “Middle Late”
- Line 367: “selected participants was”  “selected participants were”
- Line 368: referenced  registered?
- Line 389: How is significance of PLI defined?
- Equation (19): Why are w_{max} and w_{min} in equation (19) 12 and 8 Hz, instead of 14 and 6 Hz corresponding to the alpha band as in line 378?
- Line numbering missing for the entire “IPAF estimation” paragraph
- In the “IPAF estimation” paragraph”, acronyms “EOG”, “ECG” and “ICA” are used for the first time without the full terms provided.

Reviewer #2 (Remarks to the Author):

In their paper, Pathak and colleagues provide evidence that In MEG data from subjects of varying ages across the lifespan, they first demonstrate that phase-locking measures (PLV and PLI) reflecting neuronal synchrony do not change over the lifespan at the peak alpha frequency, which decreases with age, suggesting that there is a form of neural compensation occurring in the form of preserved coupling in response to natural delays in axonal transmission over age. They then show theoretically and analytically how conduction delays give rise to reductions in network synchrony using an all-to-all network modeled with Karamuto phase-oscillators, and that this reduction in synchrony can be rescued by increasing the coupling between oscillators. Then the authors model the relationship between oscillation frequency, conduction delays, and coupling on structural connectomes parcellated with cortical atlases. They demonstrate that reduced conduction speeds can be rescued with increasing neural coupling/phase locking, and that that alpha frequency suppression emerges from this compensation rather than as an age-associated pathology.

The paper is generally well-written and clear in its methods, and I have few points of critique. My primary questions and suggestions pertain to the connectome modelling approach. For one, it is unclear why the authors did not use a structural connectome atlas that contains subcortical structures such as the thalamus, given that thalamic inhibition can contribute significantly to neural dynamics. Second, the modelling of conduction delays could be made more anatomically accurate by incorporating patterns of white matter integrity disruption observed in their dataset across age, as opposed to their current approach that assumes that all connections lose conduction velocity based on Euclidean distance.

For instance, using their HCP cohort the authors could identify to what degree each white matter edge is affected by aging and introduce conduction delays proportional to distance as well as empirical age-related reduction in SC.

The paper would also be enhanced by sharing code that would enable one to reproduce the analyses.

Reviewers' comments:

Reviewer #1 (Remarks to the Author):

This manuscript by Pathak and others reported an analysis of diffusion MRI and MEG data over the lifespan, aiming to establish a theory explaining the slowing down of peak alpha frequency with age. This is an interesting goal and the authors are aiming to perform an ambitious study. However, I think that this manuscript does not fit well with this journal. I will explain my concerns below.

1. First, the manuscript is written solely from an engineering standpoint, the methods are complicated, and there is a lot of assumed knowledge that I suspect readers, and those who are interested in neural oscillation, white matter and ageing, will not have. If authors aim to publish this work in general biology/neuroscience journals, an ample revision is necessary to contextualize and relate the description with existing neuroscience knowledge. Important concepts, which are necessary to understand the scope of this work, must be explained in more detail.

Action:

Below we provide specific instances where we have sought to bring further clarity to our scientific narrative-

A. We introduce the idea of 'compensation' from a basic biological standpoint to address the reviewer's concerns about relating our work with existing neuroscience knowledge. For example, in the introduction section (Line:24-48)-

"Compensatory mechanisms play a central role in the maintenance of complex biological systems like the brain. The basic principle underlying compensatory processes is that similar activity patterns in biological networks can be achieved by multiple underlying dynamical configurations of the system. For example, neural networks can preserve circuit operation around a target set point in the face of continuous molecular turnover by undergoing synaptic modifications. Although compensation has been mostly studied at fast timescales, compensatory processes are also relevant at ultra-slow timescales such as lifespan ageing. A persistent debate in the field of ageing neuroscience centres around the question of whether functional neuromarkers of healthy ageing indicate a gradual degradation of brain structure or, the presence of compensatory reorganization mechanisms that counteract the deleterious effects of structural loss. Compensatory theories posit that certain features of brain ageing, for example, enhanced activation in specific brain regions with age are markers of compensation rather than functional decline. Behaviorally, compensatory theories of ageing are further supported by reports of age-related maintenance of cognitive domains such as language comprehension and crystallized intelligence. However, the task of classifying various markers of brain ageing as either adverse or compensatory is made difficult by the enormous complexity of the brain. Biophysically inspired computational models provide crucial mechanistic insights when purely experimental observations are insufficient in resolving competing hypotheses. Therefore, the goal of

this manuscript is two fold - first, to identify dynamical markers that remain invariant with age and second, to elucidate the operational principles that support such invariant relationships in the face of age-related structural decay.”

B. The central premise of our article is stated with greater clarity in the introduction (Line: 78-92)-

“Here, we hypothesize that PAF slowing with age is symptomatic of compensation, rather than just being a passive fallout of age-related structural deterioration. To test our claim, we evaluate the degree of neural coordination using phase synchrony measures at different regimes of alpha frequency band in resting-state MEG recordings collected from participants aged 18-88 at the Cambridge Centre for Ageing and Neuroscience (Cam-CAN) . Phase synchrony was measured by estimating Phase Locking Value (PLV)- a measure that quantifies the consistency of phase relationships between oscillators over a period of time. For each subject in the cohort we calculated PLV at three different frequency bands- Lower Alpha (LA, 6-10 Hz), Upper Alpha (UA, 10-14Hz) and another subject specific band obtained by considering a 4Hz band centred at the PAF (Subject Specific Alpha or SSA). We conjectured that while phase synchrony will be altered with age for the LA and UA bands, it will remain preserved at the SSA band. We further reasoned that the polarity of correlation between phase coupling and age (whether increasing or decreasing) in each of the three bands would hint at the underlying mechanism of frequency slowing.”

C. We provide a more layman account for domain-specific information to address the reviewer’s concerns about the manuscript being written from an engineering standpoint. For example, in the results, we describe PLV and PLI thus (Line: 135-144)-

“PLV is estimated by bandpass filtering the time-series in the frequency band of interest and then using Hilbert transform to obtain the corresponding phase time-series. Next, for each ROI pair, phase time-series are subtracted to yield phase differences which are then used to estimate consistent phase relations across time (see Methods for mathematical expression). While robust at estimating phase locking, PLV is susceptible to picking up phase correlations that may arise due to the spread of magnetic fields from one brain area to another (known as field spread). Phase correlations due to field spread are most likely to occur at zero lags. Therefore, in addition to PLV, another measure of phase locking--- Phase Lag Index (PLI), which discounts zero-lag correlations, was estimated for each participant and frequency band (See Methods and Figure 1)”

D. We append a glossary at the end to further clarify technical terminology.

2. For example, many acronyms (PAF, PLI, PLV, or IPAF) were used in this manuscript but they are not carefully described to readers who are not very familiar with EEG/MEG. It seems that authors assume that readers have prior knowledge on these terms, but I don't think that all readers have this knowledge considering the scope of this journal. Importantly, some acronyms, such as IPAF, were used without even providing the full terms when they first appeared in the text. For this reason, as a reader, I have no chance to fully understand the author's method.

Action:

A. We now provide a clearer definition of PAF in the introduction itself (Line: 52-57)-

“EEG/MEG recordings at rest are marked by prominent oscillatory activity in the alpha frequency range (8-12Hz). Numerous studies have shown that the frequency corresponding to the peak power in the alpha band (Peak Alpha Frequency or PAF), slows with normal ageing”

B. PLV and PLI are carefully described (See Point 1 C,D).

C. To improve readability we now replace IPAF (Individual Peak Alpha Frequency) with PAF (Peak Alpha Frequency) as both terminologies seem to be conveying the same meaning.

3. In addition, Level of details provided regarding methods in the Results section should be revised so that it is appropriate for the format with Results placed before Methods. In the current form, the reader must go back and forth between Results and Methods to understand what was done.

Action:

1. As mentioned above, we now explain PLV and PLI in greater detail in the result section (See point 1,D).

2. We begin the Results section with a short summary of the analysis to prime the reader for the subsequent sections (Line: 106-115)-

“We performed MEG data analysis in conjunction with computational modeling to elucidate the relationship of neural coordination associated with PAF and reorganization of white-matter with age (N = 627, 315 Females). First, we estimate phase locking - a widely used measure of neural coordination among simultaneously recorded brain signals, from eyes closed resting-state MEG data (~8 minutes/participant). Second, we theoretically demonstrate compensatory re-organization in a simplified network model of interacting alpha oscillators connected via all-to-all connection topology.

Finally, we simulate the neural dynamics generated by the cortical network spanning the entire brain whose nodes are connected by DTI based structural connectivity with realistic cortical conduction speed to gain insights into the biological mechanisms that guide the slowing of PAF and maintenance of neural coordination.”

4. The authors separately analyzed lower-alpha (LA) and upper alpha (UA). However, it is fully unclear to me why authors divided frequency bands in this way. Are there any theoretical grounds that these frequency bands have distinct physiological origins? Without providing sufficient context, analysis dividing LA and UA is fully incomprehensible to readers.

Action:

A. We now clearly state the motivation for analyzing LA and UA bands in the introduction (Line: 85-92)-

“For each subject in the cohort, we calculated PLV at three different frequency bands- Lower Alpha (LA, 6-10 Hz), Upper Alpha (UA, 10-14Hz) and another subject-specific band obtained by considering a 4Hz band centred at the PAF (Subject Specific Alpha or SSA). We conjectured that while phase synchrony will be altered with age for the LA and UA bands, it will remain preserved at the SSA band. We further reasoned that the polarity of correlation between phase coupling and age (whether increasing or decreasing) in each of the three bands would hint at the underlying mechanism of frequency slowing.”

We further justify our choice of studying LA/UA separately in the discussion (Line:297-300) -

“By studying phase coupling separately for LA, UA and SSA bands we confirm that in the presence of group variability in peak frequencies, the use of predefined frequency bands to characterize group differences in functional connectivity leads to the detection of spurious relationships”

5. Authors averaged peak alpha data across all ROIs. What are the theoretical grounds for this step? Alpha oscillation in the occipital cortex and frontal cortex may have a very different nature since they are distinct signals and have different functional properties. Averaging peak alpha from all ROIs makes it very complicated to understand the relationship with white matter tracts, which connect specific brain regions.

Action:

We thank the reviewer for raising the crucial issue of the physiological origins of alpha rhythmicity. We would like to point out that despite years of research, the biophysical mechanism underlying alpha rhythmogenesis and propagation remains intensely controversial. For example, the field is still undecided about which of the two- thalamus or cortex- is the main driver of alpha activity, with several supporting accounts existing for either position * (Halgren 2019). Even the dynamical nature of alpha

rhythms- whether filtered noise, noise buffeted deterministic oscillations or chaotic remains far from settled ******(DaSilva 1997). Therefore, several lacunae exist in our present understanding which makes it difficult to extend our findings spatially (i.e topography), and so we follow the approach of several previous studies and resort to working with average PAF. Moreover, asking specific questions about topographical organization requires individual structural connectivity (SC) matrices, which we do not possess at this moment. We think that characterizing age-related changes in the topographical organization of brain rhythms is a promising avenue for future research. Interestingly, our model allows for adding heterogeneity in connection strengths, transmission delays and natural frequencies and can be potentially extended to incorporate individual SCs for obtaining better topographical understanding.

*Halgren, Mila, István Ulbert, H el ene Bastuji, D aniel Fab o, Lorand Er oss, Marc Rey, Orrin Devinsky et al. "The generation and propagation of the human alpha rhythm." *Proceedings of the National Academy of Sciences* 116, no. 47 (2019): 23772-23782.

**Da Silva, FH Lopes, J. P. Pijn, D. Velis, and P. C. G. Nijssen. "Alpha rhythms: noise, dynamics and models." *International Journal of Psychophysiology* 26, no. 1-3 (1997): 237-249.

6. Information regarding human subjects is not sufficient for readers to understand this work. Even though the analysis was performed by using public datasets, information of human subjects (age, sex, and others) should be described in more detail.

Action:

We provide an age histogram in the supplementary section (Supplementary). We also add the sex split in the main text (Line: 106-108)-

"We performed MEG data analysis in conjunction with computational modeling to elucidate the relationship of neural coordination associated with PAF and reorganization of white-matter with age (N = 627, 315 Females)"

7. Importantly, it is not clear what is the rationale for randomly picking 200 subjects. Why 200 subjects were chosen for this analysis. If authors pick different 200 subjects, will they obtain consistent results?

Action:

A. We thank the reviewer for raising the question of reproducibility of our findings. First, we would like to point out that the initial choice of randomly selected 200 subjects out of the original 650 was made in order to save computation time as source reconstruction algorithms can be prohibitively time-consuming. However, in the time since our original submission, we have performed source reconstruction for N = 627 subjects. Besides, now we employ Minimum Norm Imaging (MNI) pipeline

as opposed to the original sLORETA, in order to further resolve confounds that may arise due to the choice of the source reconstruction algorithm. Additionally, now we utilize empty room recordings from the Cam-CAN project (made recently available) for noise covariance estimation, thereby reducing artifactual phase locking. Our findings with N=627 subjects remain qualitatively similar to what we report in the original manuscript with 200 subjects. (See updated Fig 2.). Namely, we find robust peak alpha frequency reduction and preservation of Phase locking with age at the PAF. To further allay the reviewer's concerns over the reproducibility of our findings, we repeat the analysis of Scally et. al. on a publicly available resting-state EEG dataset (SRM, N = 111, age-range: 17-71). In agreement with our findings of preservation of phase-locking by frequency slowing, Scally et. al.(2018) report the absence of group differences in Phase locking at the peak alpha frequency between young and old participants. Following Scally et. al., we estimate phase locking between EEG electrodes and estimate the frequency of maximum phase locking in the alpha band for each subject and correlate it with PAF. We find a positive correlation between PAF and peak PLI frequency, indicating a relationship between spectral power and network synchrony. (Supplementary fig compared with Fig 3. of Scally et. al. 2018).

8. Since not all readers are fully familiar with the MEG analysis pipeline, terms like “boundary element method”, “Welch periodogram method” and “Hanning window” should be explained in a more transparent way.

Action: We now describe the Welch method more transparently (Line: 430-435)-

“All subsequent analysis was performed on the resulting data. ROI-wise power spectral density was estimated using the Welch periodogram method. In accordance with the Welch method (Welch, 1967), the time series was split into 5s windows with 50 % overlap. Each window was multiplied with a tapering window (Hanning window) to suppress the contribution of spectral leakage. Discrete Fourier transform was performed to yield Fourier coefficients, which were then squared to yield spectral power.”*

However, we hope that the kind reviewer would appreciate that some of these terminologies are considerably involved, and therefore, in the interest of brevity we limit ourselves to providing citations, wherever appropriate.

For example, in the Pre-processing and Source Reconstruction section, we direct the reader to Baillet et. al.** for a discussion on Minimum Norm Imaging. Similarly, for an in-depth review of the Brainstorm pipeline, we cite Niso et. al.***

*Welch, Peter. "The use of fast Fourier transform for the estimation of power spectra: a method based on time averaging over short, modified periodograms." *IEEE Transactions on audio and electroacoustics* 15, no. 2 (1967): 70-73.

** Baillet, Sylvain, John C. Moshier, and Richard M. Leahy. "Electromagnetic brain mapping." *IEEE Signal processing magazine* 18, no. 6 (2001): 14-30.

***Niso, Guiomar, Francois Tadel, Elizabeth Bock, Martin Cousineau, Andrés Santos, and Sylvain Baillet. "Brainstorm pipeline analysis of resting-state data from the open MEG archive." *Frontiers in neuroscience* 13 (2019): 284.

9. It would be useful to have the values of the IPAF centred band as for LA and UA. By looking at Fig 2. a, it seems that they might coincide with LA for some of the age groups?

Action: We agree with the reviewer that the SSA (Subject-specific alpha) band has significant overlap with the LA (Lower alpha) band. As seen in Fig 2a, the mean PAF lies between 8-11 Hz across participants. To make this point explicit, we provide an additional figure in the supplementary section representing all three bands for each participant (Supplementary).

We also direct the reviewer to the updated Fig 2c, which further addresses the reviewer's concerns by quantifying the statistical difference in PLV/PLI means between SSA and LA/UA bands. As pointed out by the reviewer, LA and SSA bands have considerable overlap as evidenced by the non-significant difference in PLI means ($p = 0.08$). However, the PLV analysis still indicates a significant difference between SSA and LA bands, thereby proving that the SSA band is distinct from the LA band.

10. There are many errors in the manuscript which makes it difficult for me to evaluate the quality of this work. For example:

Action: We thank the reviewer for her/his astute observations and accordingly make the following corrections-

- Insert a space between a word and the first of the following pair of brackets

Corrected

- Line 77: typo "deyterminants"  "determinants"

Corrected (Line: 384)

- Line 78: typo "metstability"  "metastability"

Corrected (Line: 254)

- Line 82: typo "was"  "were"

Corrected

- Line 97: inconsistent capitalisation "Phase locking value"  "Phase Locking Value"

Corrected

- Line 101: PLI was introduced as phase lagged index in Introduction, not as “Phase Locking Index”?

Corrected, its Phase Lag Index(PLI) throughout.

- Line 136: inconsistent capitalisation “analytical solution”  “Analytical solution”

Corrected

- Line 156: ODE not previously used acronym

ODE expanded to ‘Ordinary Differential Equation’.

- Line 381: What is the SSA band?

We now explain the SSA in the introduction itself(Line: 85)-

“For each subject in the cohort we calculated PLV at three different frequency bands- Lower Alpha (LA, 6-10 Hz), Upper Alpha (UA, 10-14Hz) and another subject-specific band obtained by considering a 4Hz band centred at the PAF (Subject Specific Alpha or SSA)”

- Line 356: MRC-CBSU not previously used acronym

Corrected, MRC- Cognition and Brain Sciences Unit.

- Line 362: inconsistent capitalisation “Data selection and Source Reconstruction”  “Data selection and source reconstruction”

Corrected, the section is now called ‘Preprocessing and Source Reconstruction’

- Line 365: inconsistent capitalisation “Middle late”  “Middle Late”

Corrected

- Line 367: “selected participants was”  “selected participants were”

The whole line is Preprocessed MEG data of selected participants. Therefore, ‘was’ is appropriate. The word ‘data’ may be used both as a singular or a plural.

- Line 368: referenced  registered?

Corrected

- Line 389: How is the significance of PLI defined?

The significance threshold was kept at 0.05, which is now mentioned in the text (Line: 454).

- Equation (19): Why are wmax and wmin in equation (19) 12 and 8 Hz, instead of 14 and 6 Hz corresponding to the alpha band as in line 378?

Corrected, we use the 8-12 Hz definition of alpha to exclude participants.

However, in order to establish the robustness of our model we simulated the 6-14Hz case as well (See supplementary) and found qualitatively similar results.

- Line numbering missing for the entire “IPAF estimation” paragraph

Corrected

- In the “IPAF estimation” paragraph”, acronyms “EOG”, “ECG” and “ICA” are used for the first time without the full terms provided.

Corrected

Reviewer #2 (Remarks to the Author):

In their paper, Pathak and colleagues provide evidence that In MEG data from subjects of varying ages across the lifespan, they first demonstrate that phase-locking measures (PLV and PLI) reflecting neuronal synchrony do not change over the lifespan at the peak alpha frequency, which decreases with age, suggesting that there is a form of neural compensation occurring in the form of preserved coupling in response to natural delays in axonal transmission over age. They then show theoretically and analytically how conduction delays give rise to reductions in network synchrony using an all-to-all network modelled with Karamuto phase-oscillators, and that this reduction in synchrony can be rescued by increasing the coupling between oscillators. Then the authors model the relationship between oscillation frequency, conduction delays, and coupling on structural connectomes parcellated with cortical atlases. They demonstrate that reduced conduction speeds can be rescued with increasing neural coupling/phase locking and that alpha frequency suppression emerges from this compensation rather than as an age-associated pathology.

The paper is generally well-written and clear in its methods, and I have few points of critique. My primary questions and suggestions pertaining to the connectome modelling approach.

1. For one, it is unclear why the authors did not use a structural connectome atlas that contains subcortical structures such as the thalamus, given that thalamic inhibition can contribute significantly to neural dynamics.

Action:

Here, we would like to echo our earlier remarks in response to reviewer 1 (see point 5, R1). Indeed, thalamic inhibition has been implicated in alpha dynamics by several studies (Robinson et. al.). However, another set of studies have identified cortical origins for alpha rhythmicity (Lopes da Silva et. al.). In order to steer clear of the controversy, we present a model of large-scale alpha synchronization that leans towards a phenomenological explanation for a well-reported phenomenon in the ageing neuroscience literature at the expense of biophysical details. Here, cortical areas are represented by

autonomous limit-cycle oscillators which interact through white-matter connectivity. We intentionally overlook the genesis of the oscillatory activity (whether thalamic or cortical) and instead focus on how the underlying white-matter network may modulate individual oscillators, finding that this approach offers a simple and biologically valid explanation for the compensatory reorganization of network frequency due to slowed axonal transmission with age. In fact, we briefly weigh on this very subject in the discussion (Line: 303)-

“Therefore, in order to study the relationship of network frequency and synchrony, we reduce the large-scale white-matter network to its basic dynamical elements— conduction delays and inter-areal coupling that forms the backbone of a whole-brain connectome. Each node in the connectome is considered to be a unit amplitude limit-cycle oscillator (an idealized autonomous oscillator), described by its phase. Anatomically, each autonomous alpha oscillator can be identified with a self-sustained thalamocortical unit. or alternatively, pacemaker populations such as the infragranular and supragranular layer in V2 and V4”

That is not to say that there could not exist alternate explanations for the phenomenon. In fact, in our discussion, we cite Van Albada and colleagues, who employ a more detailed neural mean-field model of the thalamocortical system to investigate age-associated changes in EEG spectral parameters and find white-matter stabilization and regression to be a major determinant of EEG characteristics across life-span. We also cite the study of Bhattacharya et. al. which also uses a thalamocortical model. However, as we point out in the discussion, these studies overlook large-scale cortical conduction delays. Furthermore, the introduction of multiple time-delays in mean-field models makes it difficult to obtain analytical solutions and clutters the parameter space.

Thus, the question of how alpha oscillations come about in the first place and how the current model may be augmented with that information would be an interesting subject for future explorations but would require extensive developments in mathematical analysis of non-linear systems with time delays. Simply adding a thalamic node to the WBM would just produce the same effect as before without offering any fundamental insights into age-related frequency slowing as a function of thalamic contributions. To gain a better understanding of thalamic involvement in frequency slowing, a more detailed thalamic compartment (involving reticular inhibition) would need to be constructed and integrated with cortical SC. All these are excellent future directions.

* Robinson, P. A., C. J. Rennie, D. L. Rowe, S. C. O'Connor, J. J. Wright, E. Gordon, and R. W. Whitehouse. "Neurophysical modeling of brain dynamics." *Neuropsychopharmacology* 28, no. 1 (2003): S74-S79.

* Da Silva, FH Lopes, and W. Storm Van Leeuwen. "The cortical source of the alpha rhythm." *Neuroscience letters* 6, no. 2-3 (1977): 237-241.

2. Second, the modeling of conduction delays could be made more anatomically accurate by incorporating patterns of white matter integrity disruption observed in their dataset across age, as opposed to their current approach that assumes that all connections lose conduction velocity based on Euclidean distance. For instance, using their HCP cohort the authors could identify to what degree each white matter edge is affected by aging and introduce conduction delays proportional to distance as well as empirical age-related reduction in SC.

Action:

We would like to point out that the SC matrices used for whole-brain modelling are not for the same set of subjects for whom MEG analysis was performed. The SC matrices are simply intended as template white-matter networks to demonstrate the biological validity of our model. Indeed, with lifespan structural information it is possible to incorporate spatial heterogeneity in how network conduction speeds are affected with age. However, it is still not clear how conduction speeds can be directly inferred just from SC matrices since DTI adjacency matrices contain information about fibre density, and the precise relationship (if any) between fibre density and conduction speeds has not been established. More sophisticated measures like g-ratio mapping, which quantify myelination profiles in individual fibres would be required to introduce heterogeneity in conduction delays. In that case, as suggested by the reviewer, myelination levels could be reasonably estimated at the group level and delays scaled accordingly. However, extending this approach to individual participants would still be error-prone.

That said, we add the following line in the discussion to emphasize this point (Line: 394)-

“Going forward, our model may be augmented by new imaging paradigms (eg. g-ratio mapping) to introduce heterogeneity in conduction speeds.”

3. The paper would also be enhanced by sharing code that would enable one to reproduce the analyses.

Action: We thank the reviewer for the suggestion and accordingly share codes for the main functions involved in source reconstruction, 1/f removal, PLV/PLI extraction and connectome modelling.

Link:

https://bitbucket.org/cbdl/wbm_kuramotodelay/src/master/

Reviewers' comments:

Reviewer #1 (Remarks to the Author):

Authors addressed a majority of points raised in a previous review.

However, I am still not convinced about a rationale for separately analyzing lower-alpha (LA) and upper alpha (UA). I asked authors to provide theoretical justifications for such analysis, but in my view, authors do not provide sufficient justifications. This analysis choice (separating LA, UA, and subject-specific band) looks ad-hoc and as a reader I simply do not understand motivation, namely, what authors would like to understand by comparing LA, UA, and subject-specific band. I think that authors should aim to provide more explicit argument on why authors did such selection and theoretical grounds for justifying such selection. Including citations for justifying this analysis strategy will be welcome.

Minor point

lines 60-61: Author may also consider to cite:

Valdés-Hernández, P. A., Ojeda-González, A., Martínez-Montes, E., Lage-Castellanos, A., Virués-Alba, T., Valdés-Urrutia, L., & Valdes-Sosa, P. A. (2010). White matter architecture rather than cortical surface area correlates with the EEG alpha rhythm. *Neuroimage*, 49(3), 2328-2339.

Reviewer #2 (Remarks to the Author):

The authors have made considerable improvements to the interpretability and accessibility of their article, and have addressed my main concerns.

Response to reviews

Reviewer #1 (Remarks to the Author):

1. Authors addressed a majority of points raised in a previous review.

Thank you

2. However, I am still not convinced about a rationale for separately analyzing lower-alpha (LA) and upper alpha (UA). I asked authors to provide theoretical justifications for such analysis, but in my view, authors do not provide sufficient justifications. This analysis choice (separating LA, UA, and subject-specific band) looks ad-hoc and as a reader, I simply do not understand motivation, namely, what authors would like to understand by comparing LA, UA, and subject-specific band. I think that authors should aim to provide more explicit argument on why authors did such selection and theoretical grounds for justifying such selection. Including citations for justifying this analysis strategy will be welcome.

We have added theoretical justifications for splitting the alpha band in upper, lower and subject-specific alpha in the Introduction as suggested by the reviewer

Lines 89-97:

“Several cross-sectional studies have found that spectral features of on-going alpha oscillations exhibit differential effects with age in the lower and higher alpha sub-bands Babiloni et al. [2006b], Vecchio et al. [2014], Scally et al. [2018], Ga’al et al. [2010]. Moreover, the lower and higher alpha bands have been found to be functionally independent in several cognitive and perceptual paradigms Sigala et al. [2014], Kumar et al. [2020], Petsche et al. [1997], further justifying a sub-division of the alpha band. We further reasoned that the polarity of correlation between phase coupling and age (whether increasing or decreasing) in each of the three bands would aid model building by further restricting the search space of candidate mechanisms.”

In the Discussion section, we have further elaborated on the division of alpha-band into sub-bands and argue how the patterns observed in upper alpha/ lower alpha and SSA bands are vital observations in setting the parameter space of the computational model.

Lines 302-328:

“Furthermore, partitioning the alpha band into lower and higher sub-bands offers both practical and mechanistic insights into the nature of dynamical re-organization over the course of physiological ageing. By studying phase coupling separately for LA, UA and SSA bands we are able to confirm that in the presence of group variability in peak frequencies, the use of predefined frequency bands to characterize group differences in functional connectivity leads

to the detection of spurious relationships Scally et al. [2018]. Therefore, our findings caution against the use of pre-selected frequency bands for studying cognitive phenomenon Scally et al. [2018] while also highlighting the importance of considering inter-subject/condition variability in the distribution of neural oscillations Haegens et al. [2014], Quinn et al. [2021]. A similar approach of dividing alpha into a lower and higher sub-band is followed by Vecchio et al. who showed that graph-theoretic connectivity in the alpha band decreases with age in the higher alpha sub-band Vecchio et al. [2014]. Gaal et al. also study physiological ageing by performing EEG reactivity analysis by considering two sub-bands within alpha Gaal et al. [2010]. Similarly Babiloni et al. characterize age trajectories separately for alpha sub-bands and report differential effects in the lower and higher alpha frequency ranges Babiloni et al. [2006a]. Besides ageing, another group of studies have divided the alpha band to shed light on neurological disorders such as Mild Cognitive Impairment (MCI) Babiloni et al. [2014], Moretti et al. [2013], Persistent Vegetative State Babiloni et al. [2009] and Autism Spectrum Disorder (ASD) Lefebvre et al. [2018]. Most importantly, age-related trajectories in the LA, UA and SSA bands serve as the explanandum for our subsequent model. While preserved age trajectories in the SSA band may come about due to multiple underlying phenomena, the added constraints imposed by LA and UA trajectories allow us to restrict the computational model to a smaller set of candidate mechanisms (Figure 2d). In short, our computational model aimed to explain three crucial empirical observations-1) Preserved phase locking in the SSA band alongside a reduction in PAF 2) Enhanced phase-locking with age in the LA band and 3) Reduced phase-locking with age in the UA band. We argue that age-associated enhancement in conduction delays offers a parsimonious explanation for all three empirical findings.”

New citations added:

Scally, Brian, Melanie Rose Burke, David Bunce, and Jean-Francois Delvenne. "Resting-state EEG power and connectivity are associated with alpha peak frequency slowing in healthy aging." *Neurobiology of aging* 71 (2018): 149-155.

Babiloni, Claudio, Giuliano Binetti, Andrea Cassarino, Gloria Dal Forno, Claudio Del Percio, Florinda Ferreri, Raffaele Ferri et al. "Sources of cortical rhythms in adults during physiological aging: a multicentric EEG study." *Human brain mapping* 27, no. 2 (2006): 162-172.

Babiloni, Claudio, Marco Sarà, Fabrizio Vecchio, Francesca Pistoia, Fabio Sebastiano, Paolo Onorati, Giorgio Albertini et al. "Cortical sources of resting-state alpha rhythms are abnormal in persistent vegetative state patients." *Clinical Neurophysiology* 120, no. 4 (2009): 719-729.

Vecchio, Fabrizio, Francesca Miraglia, Placido Bramanti, and Paolo Maria Rossini. "Human brain networks in physiological aging: a graph theoretical analysis of cortical connectivity from EEG data." *Journal of Alzheimer's Disease* 41, no. 4 (2014): 1239-1249.

Gaal, Zsófia Anna, Roland Boha, Cornelis J. Stam, and Márk Molnár. "Age-dependent features of EEG-reactivity—Spectral, complexity, and network characteristics." *Neuroscience letters* 479, no. 1 (2010): 79-84.

Babiloni, Claudio, Claudio Del Percio, Roberta Lizio, Nicola Marzano, Francesco Infarinato, Andrea Soricelli, Elena Salvatore et al. "Cortical sources of resting state electroencephalographic alpha rhythms deteriorate across time in subjects with amnesic mild cognitive impairment." *Neurobiology of Aging* 35, no. 1 (2014): 130-142.

Moretti, Davide Vito, Giuliano Binetti, and Orazio Zanetti. "EEG upper/low alpha frequency power ratio relates to temporo-parietal brain atrophy and memory performances in mild cognitive impairment." *Frontiers in Aging Neuroscience* 5 (2013): 63.

Lefebvre, Aline, Richard Delorme, Catherine Delanoë, Frederique Amsellem, Anita Beggiato, David Germanaud, Thomas Bourgeron, Roberto Toro, and Guillaume Dumas. "Alpha waves as a neuromarker of autism spectrum disorder: the challenge of reproducibility and heterogeneity." *Frontiers in neuroscience* (2018): 662.

Kumar, Vinodh G., Shrey Dutta, Siddharth Talwar, Dipanjan Roy, and Arpan Banerjee. "Biophysical mechanisms governing large-scale brain network dynamics underlying individual-specific variability of perception." *European Journal of Neuroscience* 52, no. 7 (2020): 3746-3762.

Quinn, Andrew J., Gary GR Green, and Mark Hymers. "Delineating between-subject heterogeneity in alpha networks with Spatio-Spectral Eigenmodes." *NeuroImage* 240 (2021): 118330.

Haegens, Saskia, Helena Cousijn, George Wallis, Paul J. Harrison, and Anna C. Nobre. "Inter-and intra-individual variability in alpha peak frequency." *Neuroimage* 92 (2014): 46-55.

3. Minor point

lines 60-61: Author may also consider citing:

Valdés-Hernández, P. A., Ojeda-González, A., Martínez-Montes, E., Lage-Castellanos, A., Virués-Alba, T., Valdés-Urrutia, L., & Valdes-Sosa, P. A. (2010). White matter architecture rather than cortical surface area correlates with the EEG alpha rhythm. *Neuroimage*, 49(3), 2328-2339.

Action: Citation added. We thank the reviewer for suggesting this very relevant article.